# Structural basis for recruitment of TASL by SLC15A4 in human endolysosomal TLR signaling

Xudong Chen[1,14], Min Xie[2,14], Sensen Zhang[1,14], Marta Monguió-Tortajada[3], Jian Yin[1], Chang Liu[1,4], Youqi Zhang[5], Maeva Delacrétaz[3], Mingyue Song[1], Yixue Wang[1], Lin Dong[6], Qiang Ding[6], Boda Zhou[7], Xiaolin Tian[8], Haiteng Deng[8], Lina Xu[9], Xiaohui Liu[9], Zi Yang[10], Qing Chang[10], Jie Na[2], Wenwen Zeng[2], Giulio Superti-Furga[11,12], Manuele Rebsamen[3] & Maojun Yang[1,13] ✉

Toll-like receptors (TLRs) are a class of proteins that play critical roles in recognizing pathogens and initiating innate immune responses. TASL, a recently identified innate immune adaptor protein for endolysosomal TLR7/8/9 signaling, is recruited by the lysosomal proton-coupled amino-acid transporter SLC15A4, and then activates IRF5, which in turn triggers the transcription of type I interferons and cytokines. Here, we report three cryo-electron microscopy (cryo-EM) structures of human SLC15A4 in the apo monomeric and dimeric state and as a TASL-bound complex. The apo forms are in an outward-facing conformation, with the dimeric form showing an extensive interface involving four cholesterol molecules. The structure of the TASL-bound complex reveals an unprecedented interaction mode with solute carriers. During the recruitment of TASL, SLC15A4 undergoes a conformational change from an outward-facing, lysosomal lumen-exposed state to an inward-facing state to form a binding pocket, allowing the N-terminal helix of TASL to be inserted into. Our findings provide insights into the molecular basis of regulatory switch involving a human solute carrier and offers an important framework for structure-guided drug discovery targeting SLC15A4-TASL-related human autoimmune diseases.

The innate immune response is the body's first line of defense against pathogen invasion through the recognition of the pathogen-associated molecular pattern (PAMP) via the host's pattern recognition receptors (PRRs). The induction of a series of cytokines and mediators is followed to counter the infectious agents[1]. Toll-like receptors (TLRs) are an important class of PRRs that are localized on the cell membrane or endosomes/lysosomes and play a key role in the innate immune response[2,3]. Among the dozen human TLR members, TLR7/8/9, which is located on endosomes/lysosomes, can be activated by the recognition of RNA viruses and DNA viruses such as

influenza virus, SARS-CoV-2 and Herpes simplex virus-2[4–7]. In particular, TLR7 and TLR9 are highly expressed on plasmacytoid dendritic cells (pDCs), the main source of type I interferon[8]. In response to a wide array of viruses, pDCs engage TLR7 or TLR9 and rapidly produce type I interferon to mobilize antiviral immunity[2,4,8,9].

Like other innate immune signaling pathways, activation of the TLR7/8/9 pathway is dependent on adaptors for signal transduction[4,7,10]. Though Myd88 has been characterized since a long time[3], a key adaptor protein was only recently identified and named TASL (**T**LR **A**daptor interacting with **S**LC15A4 on the **L**ysosome)[11].

Solute carrier family 15 member 4 (SLC15A4, also known as PHT1) is one of four members of the SLC15 family, which belongs to the proton-coupled amino-acid/oligopeptide transporter (POT) family and the major facilitator superfamily (MFS). SLC15A1 (PepT1) and SLC15A2 (PepT2) transport di- and/or tripeptide across cell membranes, while SLC15A3 and SLC15A4 are thought to transport histidine, oligopeptides, and peptidoglycans across the lysosome membrane[12–18]. In addition to their function in transport, SLC15 family proteins also participate in the innate immune response.

Specifically, SLC15A4 is reported to mediate the recruitment of TASL to the lysosome, and the knockout of either SLC15A4 or TASL in the THP1 monocyte-like cell line blocks the TLR7/8/9 signaling pathway by impairing IRF5 activation and downstream responses[11]. Similar to the adaptors of other innate immune pathways (STING, MAVS, TRIF)[19–22], TASL also contains a LxIS motif that can be phosphorylated by upstream kinases. The transcription factor IRF5 is then phosphorylated and activated through the interaction with the pLxIS motif of TASL, which triggers the expression of downstream inflammatory factors and type I interferon. Furthermore, a large number of studies have shown that along the signaling pathway from the upstream receptor (TLR7/8/9), to the intermediate adapter (SLC15A4/TASL) and the downstream transcription factor (IRF5), the abnormal function of any of those components may lead to autoimmune diseases, such as systemic lupus erythematosus (SLE) and rheumatoid arthritis (RA)[23–31]. Therefore, an in-depth study of the molecular mechanism of this pathway is of great significance for understanding antiviral innate immunity and autoimmune disorders. However, in contrast to the other PRR pathways such as the well-studied interaction between STING and TBK1[32–35] in the ER/Golgi, the mechanism of SLC15A4-mediated TASL recruitment remains unclear, limiting our understanding of the endolysosomal TLRs pathway and the related pathology.

In the present study, we solved the cryo-EM structures of human SLC15A4 in the outward-facing apo state (monomeric and dimeric) as well as in the inward-facing TASL-bound state. The structure of two human disease-relevant proteins, key for the TLR7/8-IRF5 innate immune pathway allows us to understand the molecular basis for this important signaling module. Our results revealed the binding interface between SLC15A4 and TASL and the conformational changes of SLC15A4 upon TASL binding. Moreover, this suggests potential inhibition mechanisms of SLC15A4 by small-molecule inhibitors. Together with biochemical and cellular studies, our findings provide a structural basis for drug discovery targeting SLC15A4- or TASL-related human autoimmune diseases.

## Results

### Structure determination of human SLC15A4/TASL complex

To gain insights into the mechanism of SLC15A4−TASL-mediated TLR signaling, we initially tried to obtain the human SLC15A4/TASL complex by overexpressing and purifying human SLC15A4 and TASL separately. Unfortunately, this attempt failed due to aggregation and misfolding of TASL (Supplementary Fig. 1a, b), although the purification of human full-length SLC15A4 was successful (Supplementary Fig. 1c, d). Confocal microscopy images revealed that TASL derived from the weak promoter displayed a normal expression pattern and co-localized with SLC15A4, but aggregated when using a strong promoter (Supplementary Fig. 1e).

We next sought to design appropriate TASL truncations to avoid aggregation. To this end, RoseTTAFold[36] and PONDR[37] were utilized to facilitate the prediction of structure and disorder regions and constructed five plasmids to test the ability of these TASL portions to bind SLC15A4. The first four constructs retained the N-terminal sequences [TASL(1–30), TASL(1–20), TASL(1–215), TASL(1–20&207–301)] and the fifth plasmid contained an MBP tag at the C-terminal of TASL to avoid aggregation and misfolding (Supplementary Fig. 2a). Confocal microscopy images of HEK293T cells showed that three candidates,

TASL(1–30), TASL(1–20), and TASL(1–20&207–301) co-localized with SLC15A4 (Supplementary Fig. 2b), consistent with our previous study that showed TASL interacts with SLC15A4 via its N-terminus[11]. Since the TASL(1–30)-EGFP truncation was more flexible according to the prediction of RoseTTAFold[36] (Supplementary Fig. 2c), we chose TASL(1–20&207–301) and TASL(1–20)-EGFP truncations as the TASL expression plasmids for further study.

The co-expression of C-terminal Flag-tagged TASL(1–20&207–301) and C-terminal strep-tagged SLC15A4 in HEK293F cells yielded homogeneous and abundant complexes (Supplementary Fig. 2d). However, subsequent cryo-EM analyses of this protein complex reconstituted in LMNG micelles encountered difficulties since no prominent density signal of TASL was detected in the 2D class averages (Supplementary Fig. 2e). This is likely due to the high flexibility of TASL(1–20&207–301), which led to the failure of our attempts at 3D reconstruction of this protein complex (Supplementary Fig. 2f). Notably, the co-expression of the C-terminal strep-tagged TASL(1–20)-EGFP and C-terminal Flag-tagged SLC15A4 gave rise to a clearly observed EGFP signal (Supplementary Fig. 2g–i) that served as a rigid body to facilitate 3D alignment. Finally, we resolved SLC15A4/TASL(1–20)-EGFP complex at an overall resolution of 3.03 Å, which was of sufficient quality to allow confident modeling of the SLC15A4 and the N-terminal region of TASL (Supplementary Fig. 3). This method for the construction of a rigid-fused-EGFP on TASL allowed us to determine the orientation of the particles in 3D reconstruction and was a suitable strategy for the structural determination with the small molecular weight proteins.

### Overall structure of the human SLC15A4/TASL complex

The human SLC15A4/TASL complex occupies a volume of $60 \times 55 \times 45$ Å$^3$, with a stoichiometry of 1:1 for SLC15A4 and TASL (Fig. 1a, b). The architecture of SLC15A4 lacks a bulky soluble domain and the TM domain contains 12 TMs that mimic the canonical major facilitator superfamily (MFS) fold[38], with TM1 to TM6 forming the N-terminal bundle and TM7 to TM12 forming the C-terminal bundle (Fig. 1b). Notably, SLC15A4 consists of a luminal domain intervening TM9 and TM10 that harbors two β strands, which is much smaller than those of SLC15A1 and SLC15A2, that contain more than ten β strands in the extracellular domain (ECD)[16,17]. Both the N- and C-termini of SLC15A4 face toward the cytoplasmic side. SLC15A4 contains a lysosomal targeting motif in its N-terminal 1–31 residues. This was however too flexible to be visible in the final cryo-EM representation. The overall structure of SLC15A4 adopted an inward-facing conformation, with the transmembrane domain (TMD) region widening a cleft toward the cytosol to accommodate the insertion of a short α helix from the N-terminus of TASL (Fig. 1a–c).

Furthermore, we observed a significant electron density outgrowth in the lysosomal lumen around residue SLC15A4$^{N436}$ (Supplementary Fig. 4a), suggesting that SLC15A4 was N-glycosylated, probably to protect itself from the low pH environment and proteolytic activity of the lysosome. These findings are consistent with the results of sodium dodecyl-sulfate polyacrylamide gel electrophoresis (SDS−PAGE) analysis during protein purification (Supplementary Fig. 1d) and mass spectrometry (MS) analysis (Supplementary Fig. 4b). In line with this, mutagenesis of residue SLC15A4$^{N436}$ affected the lysosomal localization of SLC15A4 (Supplementary Fig. 4c), implying an essential role of SLC15A4$^{N436}$ glycosylation in SLC15A4 trafficking and localization.

### Binding interface between human SLC15A4 and TASL

A solvent-accessible vestibule, formed mainly by TM1, TM2, TM8, and TM10 of SLC15A4, extends from the intracellular, cytosolic region into the TMD. At the deep end of this vestibule, the binding pocket of SLC15A4 is negatively charged (Fig. 1d left). The N-terminal helix (residues 1–14) of TASL protrudes into the binding pocket of SLC15A4 by approximately 25 Å and entertains a number of interactions,

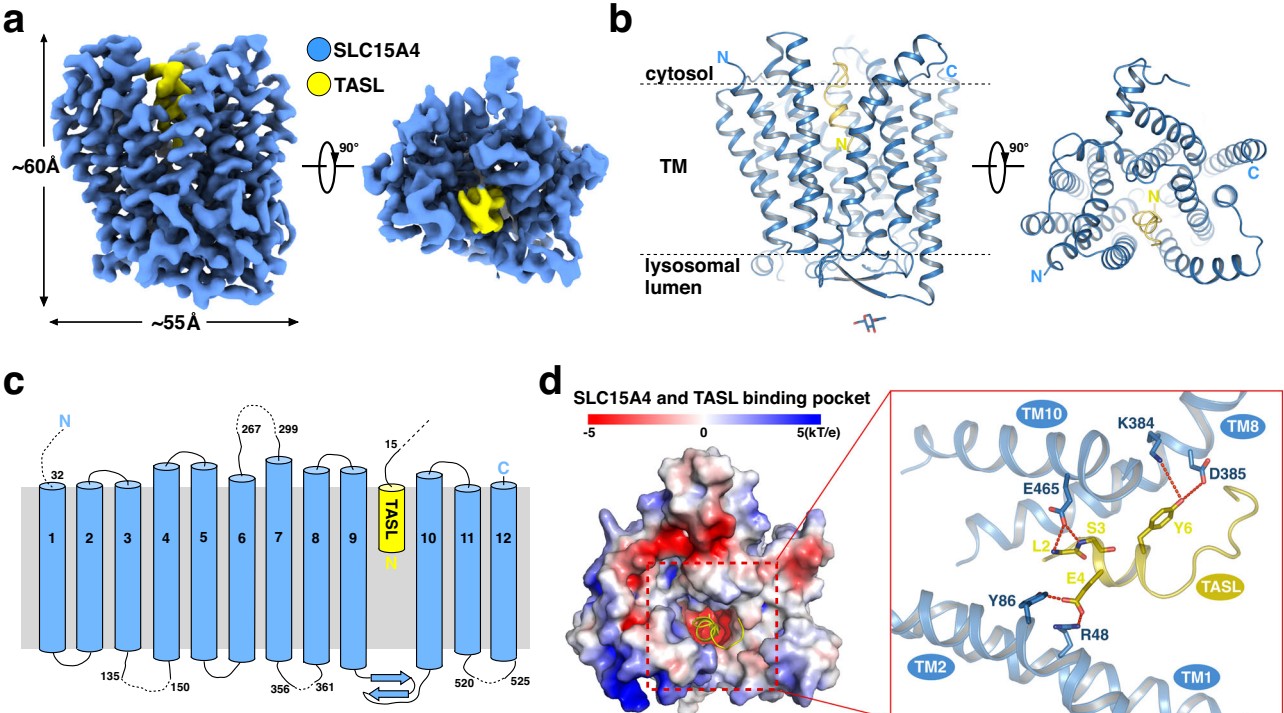

**Fig. 1 | Overall structure of human SLC15A4/TASL complex. a, b** 3.0-Å-resolution cryo-EM map (**a**) and ribbon model (**b**) of human SLC15A4-TASL complex. SLC15A4 and TASL are colored blue and yellow, respectively. The N-436 glycosylation is shown as sticks. **c** Topology diagram of human SLC15A4/TASL(1–20)-EGFP complex. TM helices and beta strands of SLC15A4 and the N-terminal sequence of TASL are schematically illustrated. The SLC15A4 and TASL are colored blue and yellow, respectively. **d** Left: Heatmap of the surface electrostatics of human SLC15A4 shows a TASL-binding pocket. TASL is shown as a yellow cartoon. Right: Close-up view of the binding interface between SLC15A4 and TASL. Potential hydrogen bonds are indicated with red dotted lines.

enabling it to form a stable TASL-SLC15A4 module. The amine nitrogen atoms of the main chains of TASL$^{L2/S3}$ were coordinated by SLC15A4$^{E465}$ via hydrogen bonds, while the side chains of TASL$^{E4}$ interacted with SLC15A4$^{R48/Y86}$. Similarly, the side chain of TASL$^{Tyr6}$ was coordinated by the side chains of SLC15A4$^{K384/D385}$ (Fig. 1d right).

To validate that these residues identified in the complex structure enable essential binding within the TMD, we performed an in vivo fluorescence co-localization assay by transiently expressing the two proteins containing the interface mutations in HEK293T cells. Substituting residues TASL$^{E4/Y6}$ to alanine in human TASL either abolished or weakened the SLC15A4-TASL interaction in vivo, whereas TASL$^{S3A}$ mutation did not affect the co-localization of the two proteins (Fig. 2a), most likely due to the fact that TASL$^{S3}$ interacts with SLC15A4 via the main-chain portion. For mutations of SLC15A4, confocal microscopy images revealed that the SLC15A4$^{E465K}$ mutation abolished the binding between the two proteins (Fig. 2b), consistent with our previous study[11], highlighting the role of this conserved glutamate in TASL binding and function (Supplementary Fig. 5). In contrast, the other two single mutants of SLC15A4$^{R48A\ or\ K384A}$, retained the ability to co-localize with TASL (Fig. 2b), suggesting a single mutation around these two residues did not completely disrupt the interaction network.

To further verify the binding mode between SLC15A4 and TASL, we next performed co-immunoprecipitation (Co-IP) assays after coexpressing SLC15A4 and TASL proteins in HEK293T cells. Consistent with the fluorescence co-localization assay in cells, the pull-down results in vitro also showed that TASL$^{E4K}$ and SLC15A4$^{E465K}$ mutations abolished the SLC15A4-TASL interaction whereas TASL$^{Y6A}$ diminished the SLC15A4-TASL interaction (Fig. 2c, d).

Given that human TLR7−9 signaling functions primarily in immune cells and that TASL is also predominantly expressed in immune cells, we next examined the TASL-SLC15A4 binding mode in THP1 cells through cell-based functional assay. THP1 TASL-null cells or SLC15A4-

null cells were generated by lentiviral transduction of gene-targeting sgRNA and Cas9. The rescue functional study of TASL/SLC15A4 wild-type or mutants in knockout cells is used to verify the binding interface between TASL and SLC15A4. The Western blot results showed that our knock-out cell lines and rescue cell lines were constructed successfully (Fig. 2e, f). Further, the cell-based functional assay revealed that R848 treatment led to robust expression of IL-8 in the cells that expressed wild-type TASL, but not in cells that expressed the TASL$^{E4K}$ and TASL$^{Y6A}$ mutants (Fig. 2e). The reciprocal assays were performed in THP1 SLC15A4-nulls cells, and the results demonstrated that SLC15A4$^{E465K}$, SLC15A4$^{R48A,}$ and SLC15A4$^{K384A}$ mutations abolished or diminished the R848-stimulated IL-8 expression (Fig. 2f). To further assess the functional effect of these mutations, we monitored TLR7/8-induced IRF5 activation as well as TNF, CCL2 and IFNβ induction in independently generated THP1 lines (Supplementary Fig. 6a, d). Similarly, we found that TASL$^{E4K}$, TASL$^{Y6A}$, SLC15A4$^{E465K}$, and SLC15A4$^{R48A}$ mutations abolished or reduced TLR7/8-induced TNF and CCL2 secretion as well as IFNβ expression (Supplementary Fig. 6b, c, e, f). These effects correlated with the reduction in R848-induced IRF5 activation, monitored by its phosphorylation, while, as expected, NF-kB and MAPK signaling was not affected as these proceed independently of the SLC15A4-TASL complex (Supplementary Fig. 6g, h)[11]. Collectively, both the in vivo and the in vitro assays confirmed the observations from the complex structure and showed that disrupting the TMD interactions of the SLC15A4/TASL module results in complex instability, which further affects IRF5 activation and downstream induction of proinflammatory cytokines and type I IFN in human immune cells.

### Apo SLC15A4 adopts an outward-facing conformation

To further illustrate the mechanism of SLC15A4-mediated TASL recruitment and investigate conformational changes associated with TASL engagement, we next resolved the structure of apo SLC15A4.

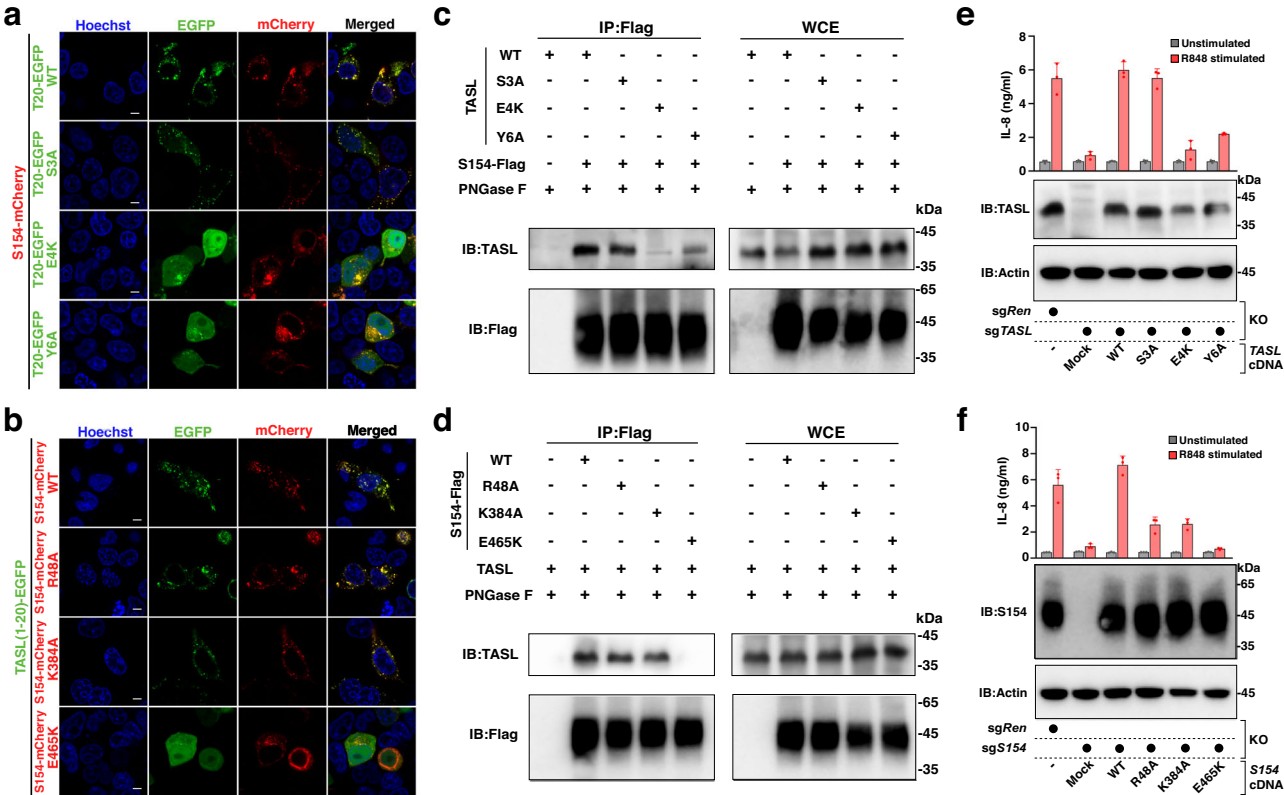

**Fig. 2 | TASL-SLC15A4 interaction is necessary for endolysosomal TLR signaling. a, b** Confocal microscopy images of HEK293T cells co-transfected with SLC15A4-mCherry and TASL(1–20)-EGFP (WT or mutations) plasmids (**a**) or TASL(1–20)-EGFP and SLC15A4-mCherry (WT or mutations) plasmids (**b**). Localization of SLC15A4 and TASL are shown. Data are representative of three biological independent experiments. S154-mCherry, SLC15A4-mCherry. Scale bars, 5 µm. **c, d** Mutations of interface residues in either TASL (**c**) or SLC15A4 (**d**) abolished or diminished the TASL-SLC15A4 interaction. Data are representative of two biological independent experiments. S154, SLC15A4. WCE, whole-cell extract. **e, f** Mutations of interface residues in both TASL (**e**) and SLC15A4 (**f**) abolished or diminished the R848-stimulated IL-8 expression in THP1 cells. THP1 TASL-null cells were reconstituted with TASL wild type or mutants (**e**), whereas THP1 SLC15A4-null cells were reconstituted with SLC15A4 wild type or mutants (**f**). Graphs show mean ± s.d. of stimulation replicates (*n* = 3 biological replicates) from one experiment representative of two independent experiments. S154, SLC15A4. Source data for relevant information are provided as a Source Data file.

Human full-length SLC15A4 was overexpressed in mammalian cells using a twin-strep-tag and subsequently reconstituted in LMNG/CHS micelle buffer (Supplementary Fig. 1c, d). The purified human SLC15A4 was subjected to cryo-EM structural analysis (Supplementary Fig. 7). Interestingly, unlike the monomeric SLC15A4/TASL complex case, cryo-EM data processing identified both the monomeric and the dimeric SLC15A4 conformations (Supplementary Fig. 7c). Although the structure of monomeric SLC15A4 was too difficult to resolve due to the low molecular weight, we successfully resolved the structure of dimeric SLC15A4 in the apo state at an overall resolution of 2.93 Å. The cryo-EM map of the dimeric SLC15A4 in the apo state showed well-resolved densities and was sufficient to build successfully most of the protein regions (Supplementary Fig. 7g).

The SLC15A4 forms a homodimer with overall dimensions of approximately 70 Å in height and 110 Å in width. The TM domain of each protomer adopts an MFS fold and exhibits an outward-facing conformation, toward the lysosomal lumen. In addition, we observed four cholesterol molecule densities within the dimer interface of SLC15A4 (Fig. 3a, b), which was confirmed by Liquid Chromatography with tandem mass spectrometry (LC-MS-MS) analysis (Supplementary Fig. 8). The SLC15A4 dimer interface was mainly composed of the TM9 and TM12 from the opposing protomer, and was stabilized by a number of interactions including the π−π interaction via F417−F417′, the hydrogen bonds interactions via N428-N428′, as well as substantial hydrophobic interactions. In addition, we found that cholesterol also plays a role in maintaining the dimerization process, for example by engaging polar interactions with K425 of TM9 (Fig. 3c). Sequence

alignment showed that the key residues in forming the SLC15A4 dimer interface, such as F417 and N428, and the cholesterol-engaging residue K425 are not conserved in human SLC15 family proteins (Supplementary Fig. 5), indicating that the ability of SLC15A4 to dimerize and engage cholesterol may be unique in SLC15 family.

Given that the dimeric architecture of apo-state SLC15A4 stands in striking contrast to the previously reported monomeric POT family transporters[16,17,39], we then performed an in vivo fluorescence resonance energy transfer (FRET) assay to verify the dimeric functional state of apo SLC15A4. We co-expressed SLC15A4-GFP and SLC15A4-mCherry in HEK293T cells and measured the FRET efficiency. The FRET assay confirmed the existence of a conformation of SLC15A4 in vivo allowing for homotypic proximity (Supplementary Fig. 9). Our FRET data are consistent with our cryo-EM structure, and strongly support the notion that the SLC15A4 dimers observed in the structure, can occur also in vivo. Interestingly, endolysosomes play an important role in cholesterol metabolism and trafficking[40]. The dimeric SLC15A4 may perform a specific function and the cholesterol plays an essential role in the dimer formation of SLC15A4.

## The monomeric SLC15A4/ALFA_Nanobody complex in the apo state

The aforementioned apo state SLC15A4 in the monomeric conformation was hard to resolve, largely due to the low molecular weight and the lack of cytosolic features. This prompted us to consider using SLC15A4-conjugated nanobodies to address this issue. Our attempts to derive nanobodies binding SLC15A4 were confronted with great

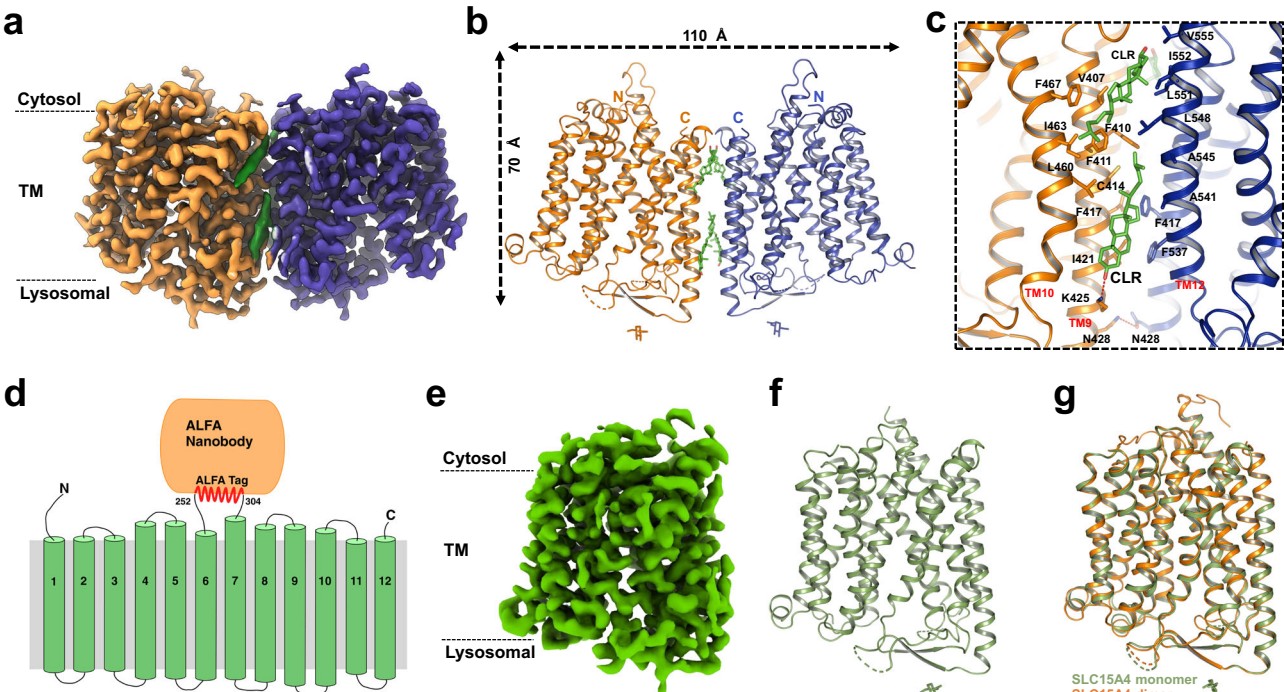

**Fig. 3 | Overall structure of human SLC15A4 in apo state. a, b** 2.9-Å-resolution cryo-EM map (**a**) and ribbon model (**b**) of human SLC15A4 dimer in the apo state. The two protomers of SLC15A4 are colored orange and blue, respectively. The bound cholesterol molecules are colored green. **c** Close-up view of the dimer interface of SLC15A4. The side chains of the indicated interface residues are shown. **d** Topology diagram of the designed SLC15A4-ALFA-Nanobody construct for resolving the SLC15A4 monomer structure. The loop (253–303) between TM6 and TM7 of SLC15A4 was replaced by an ALFA tag. The TMs, the ALFA tag, and the ALFA nanobody are colored green, red, and orange, respectively. **e, f** 3.0-Å-resolution cryo-EM map (**e**) and ribbon model (**f**) of human SLC15A4 monomer in the apo state. SLC15A4 is colored green and the N-436 glycosylation is shown as sticks. **g** Structural comparison between the monomer structure (green) and the dimer structure (orange) of human SLC15A4 in the apo state.

difficulty, so we decided to use a recently reported ALFA tag and the associated ALFA_Nanobody to facilitate the structure determination[41]. Given that the loop region between TM6 and TM7 of SLC15A4 was not visible in our structure and that a previous report showed that deleting the same loop does not affect the function the SLC15A4[11], we replaced a portion of this loop region of SLC15A4 (residues 253–303) with an ALFA tag in order to limit the movement of the bound ALFA_Nanobody (Fig. 3d). By coexpressing the ALFA tagged SLC15A4 and the ALFA nanobody in HEK293F cells followed by purification, we obtained the SLC15A4/ALFA_Nanobody complex for further cryo-EM structural analyses. Similarly as the SLC15A4 in apo state, the SLC15A4/ALFA_-Nanobody complex also contains both monomeric and dimeric conformations (Supplementary Fig. 10a). In addition, the density of ALFA_Nanobody from the 2D averages indicated that the SLC15A4/ALFA_Nanobody complex was successfully formed (Supplementary Fig. 10b).

Unlike other conventional nanobodies that have a larger binding interface, the ALFA_Nanobody bound to the short ALFA tag in the final complex is flexible and therefore the density of the ALFA nanobody in the final structure remains obscure. Nevertheless, the signal provided by the bound ALFA_Nanobody was very helpful for the 3D alignment of the particles, allowing us to finally resolve the structure of SLC15A4 in the monomeric state with a resolution of 3.25 Å. Our results demonstrate the effectiveness of this method for resolving proteins like SLC15A4, which has a small molecular weight and is composed almost entirely of transmembrane helices, thus eliminating the high cost, low success rate, and long lead time of specific antibody construction. The monomeric SLC15A4 in the apo state also adopts an outward-facing conformation, exposed to the endolysosomal lumen (Fig. 3e, f), and a comparison with the dimerized structure shows that there are no essential differences between the two structures (Fig. 3g).

Notably, differently to the SLC15A4 in the apo state and the SLC15A4/ALFA_Nanobody complex, SLC15A4 in the TASL-bound state only exhibits the monomeric conformation (Supplementary Fig. 3c). To demonstrate this difference, we superimposed two opposing SLC15A4 monomers in TASL-bound states with the outward-facing dimeric SLC15A4 structure, which yielded a pseudo-dimeric SLC15A4-TASL conformation. Structural comparison between SLC15A4 in the apo state and TASL-bound state revealed that TM9 and TM12, the two TM helices playing an essential role in forming the SLC15A4 dimer interface, bent towards the dimer interface. Notably, the distal end of TM12 moved around 4 Å and was in close proximity to the loop between TM9 and TM8, thus inducing a spatial clash within this area and further destabilizing this pseudo-dimeric SLC15A4-TASL conformation (Supplementary Fig. 11). This supports the hypothesis that TASL binding is associated with a transition between the outward-facing, endolysosomal lumen-exposed dimer and the inward-facing, cytosolic-exposed monomer.

**Conformational change of human SLC15A4 during TASL binding**
The current structures of SLC15A4 in the outward-facing apo state and TASL-bound state are now available to promote investigations into the mechanism by which SLC15A4 undergoes conformational changes to recruit TASL. The SLC15A4 in the outward-facing state possesses a solvent cleft towards the lysosomal lumen to bind possible substrates, whereas the cytosol side was sealed by the close packing of the cytosolic ends of TMs 1, 4, 5, 10, and 11 (Fig. 4a, b). The narrow exit route was closed by several interaction networks between N- and C-bundle helices, including SLC15A4[D181-R102], SLC15A4[Q485-D181-Y477], and SLC15A4[Y199-S469]. During TASL binding, helices 1, 4 and 5 from the N-bundle and helices 10 and 11 from the C-bundle move in opposite directions to enlarge the cytosol cleft to further accommodate the binding of TASL

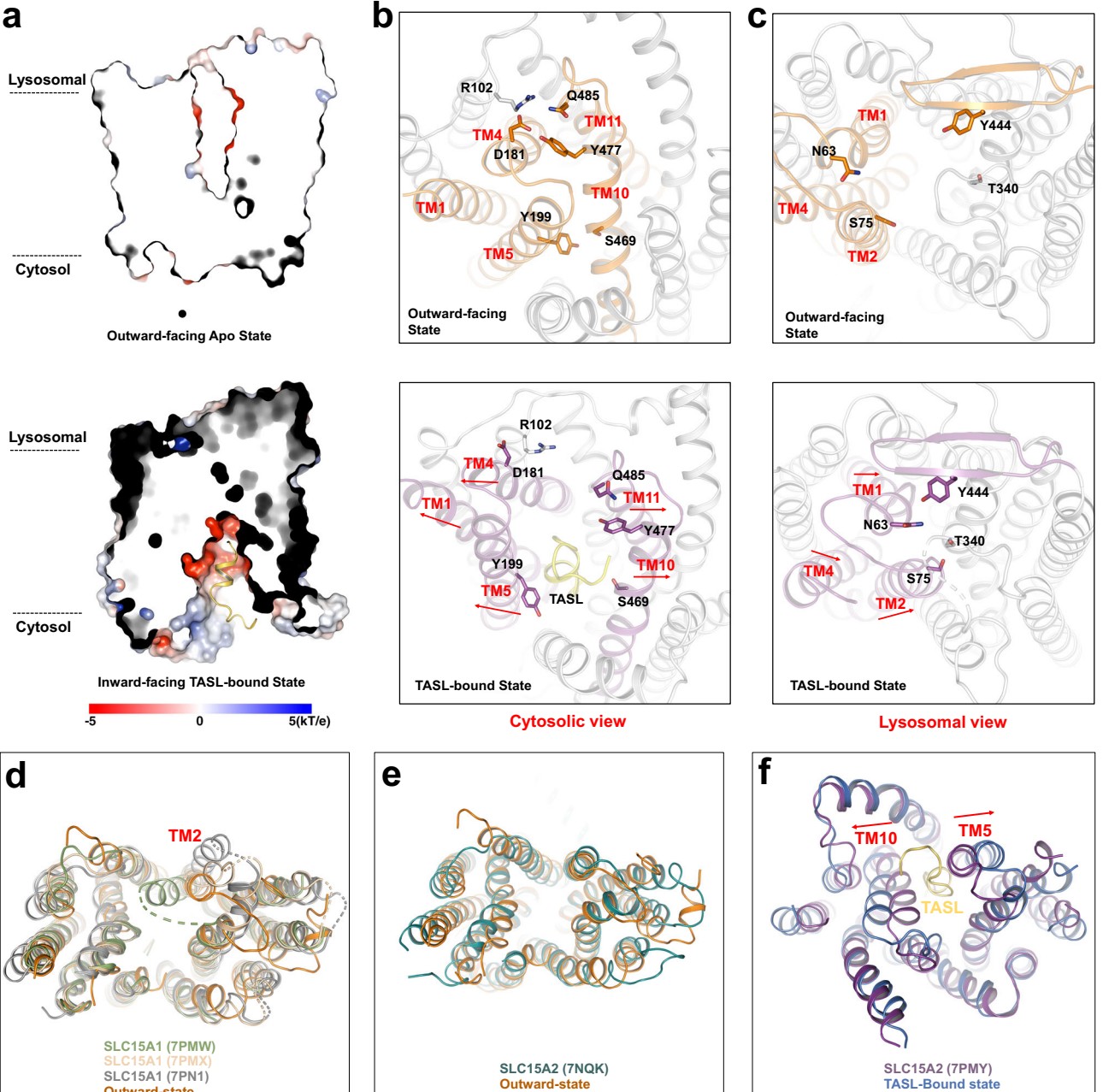

**Fig. 4 | Conformational changes of SLC15A4 during TASL binding. a** Heatmap of the surface electrostatics of SLC15A4 in the outward-facing apo state and the TASL-bound state. The TASL is shown as a cartoon model. **b**, **c** Close-up views of the movement of indicated TM helices and residues during the conformational change from an outward-facing state to a TASL-bound state in cytosolic view (**b**) and lysosomal view (**c**). The red arrows indicate the movement of TM helices and residues. **d** Structural comparison of human SLC15A4 in its outward-facing state

(orange) and three SLC15A1 models [PDB accession codes: 7PMW (green), 7PMX (wheat), and 7PN1 (gray)]. **e** Structural comparison of human SLC15A4 in the outward-facing state (orange) and SLC15A2 model [PDB accession codes: 7NQK (teal blue)]. **f** Structural comparison of human SLC15A4 in the TASL-bound state (blue) and SLC15A2 in the substrate-bound state model [PDB accession codes:: 7PMY (purple)].

(Fig. 4b). Meanwhile, the movement of helices 1, 2 and 4 induced the closure of the intra-lysosomal gate, which is open in the outward-facing state. The intra-lysosomal gate was constricted through the hydrogen-bond interactions via N63-Y444 and S75-T340 (Fig. 4c).

The transition from the outward-facing state to the TASL-bound state of SLC15A4 was largely facilitated by a large movement of the N-bundle helices and a small movement of the C-bundle helices. The conformational change of SLC15A4 was analogous to the reported POT family (SLC15A1 and SLC15A2)[16,17], but with striking differences: (a) the structure of SLC15A4 in the apo state and the three reported structures of SLC15A1 are all in the outward-facing conformation and share

similar TM helices packing mode except that TM2 of SLC15A1 varies in a different state (Fig. 4d), signifying a critical role of TM2 in substrate binding of SLC15A1; (b) the apo SLC15A4 and SLC15A2 are in the same outward-facing conformations with an RMSD of 1.242 Å over 294 Cα atoms and the structural difference is mainly associated with the C-bundle helices (Fig. 4e). (c) The inward-facing SLC15A2 in the substrate-bound state exhibits a similar conformation to SLC15A4 in TASL-bound state, except that TM5 and TM10 in the SLC15A4 TASL-bound state adopt a large cleft towards the cytosol (Fig. 4f) to facilitate accommodation of the adaptor protein TASL, which is much bigger than that of the small substrate, dipeptide Ala-Phe, in SLC15A2.

As the outward-facing conformation of SLC15A4 is incompatible with a stable interaction with TASL, the alternative conformations of the two forms, outward-facing and inward-facing, suggest several pharmacological intervention modes. Chemical compounds that bind to the outward-facing conformation on the lysosomal lumen side, may hinder the transition to an inward-facing TASL-engaging conformation. Such compounds would be effective inhibitors of the pathway as they would lead to the destabilization of TASL. Indeed, in the accompanying paper (Boeszoermeny et al.[42]), we describe the identification of such a compound as a strong inhibitor of SLC15A4 signaling, providing a valuable validation of the regulation model of the SLC15A4/TASL complex.

## Discussion

In this study, we resolved structures of the human SLC15A4 in the apo state (monomeric and dimeric), and TASL-bound state and uncovered the mechanism by which SLC15A4 functions as an adaptor to mediate the recruitment of TASL. Combined with previous results, we propose a mechanistic model of the SLC15A4/TASL-mediated endolysosomal TLRs signaling pathway. During the recruitment process of TASL from the cytosol to the lysosome mediated by SLC15A4, the homodimeric SLC15A4 undergoes a depolymerization process accompanied by a conformational change from an outward-facing state to an inward-facing state, leading to the opening of the TASL-binding pocket and the subsequent binding to TASL. After the formation of the SLC15A4−TASL module, activation of TLR7−9 by their respective ligands triggers downstream kinase activation leading to TASL phosphorylation, presumably by IKKβ. The transcription factor IRF5 is then recruited by TASL via the interaction with the pLxIS motif and is also phosphorylated. Once phosphorylated, IRF5 forms homodimers and enters the nucleus to perform transcription factor functions, resulting in the production of inflammatory cytokines and induction of antimicrobial responses. A selective inhibitor should be able to block the formation of the SLC15A4-TASL module by locking SLC15A4 in the outward-facing conformation, thus interrupting the endolysosomal TLR signaling pathway and preventing downstream inflammatory responses (Fig. 5) (see the accompanying paper by Boeszoermeny et al.[42]).

The SLC15A4-TASL complex is the key component of the lysosomal TLR signaling, and sequence alignment of SLC15A4 and TASL in different species indicates that SLC15A4 and TASL homologs share a high identity in mammals. Furthermore, all the residues that play critical roles in maintaining the SLC15A4-TASL-binding interface are perfectly conserved (Supplementary Fig. 12), suggesting that the complex represents a two-component functional module.

Previous studies reported that the aberrant activation of the TLR pathway contributes to the development of autoimmune diseases, such as SLE and RA[23–26]. Given that SLC15A4-mediated TASL recruitment plays a critical role in TLR signaling, and both of the two proteins were reported to be involved in SLE[27,43–45] as a result of overexpression or overactivation, a strategy for treating SLE could focus on blocking or reducing the interaction between SLC15A4 and TASL based on our structural findings. Excitingly, such a drug candidate is identified in the accompanying paper (Boeszoermeny et al.[42]), and more drugs may be discovered in the future. Moreover, recent studies have shown that loss-of-function mutations in TLR7 are closely related to severe cases of COVID-19[5,6], so our structural information of this pathway could be of great significance for the treatment of COVID-19 as well by assisting the identification of activating agents.

In conclusion, our findings reveal the molecular mechanism of SLC15A4-mediated TASL recruitment, which not only provides new insights into the structural features of SLC15 and the POT families but also expands our understanding of human endolysosomal TLR

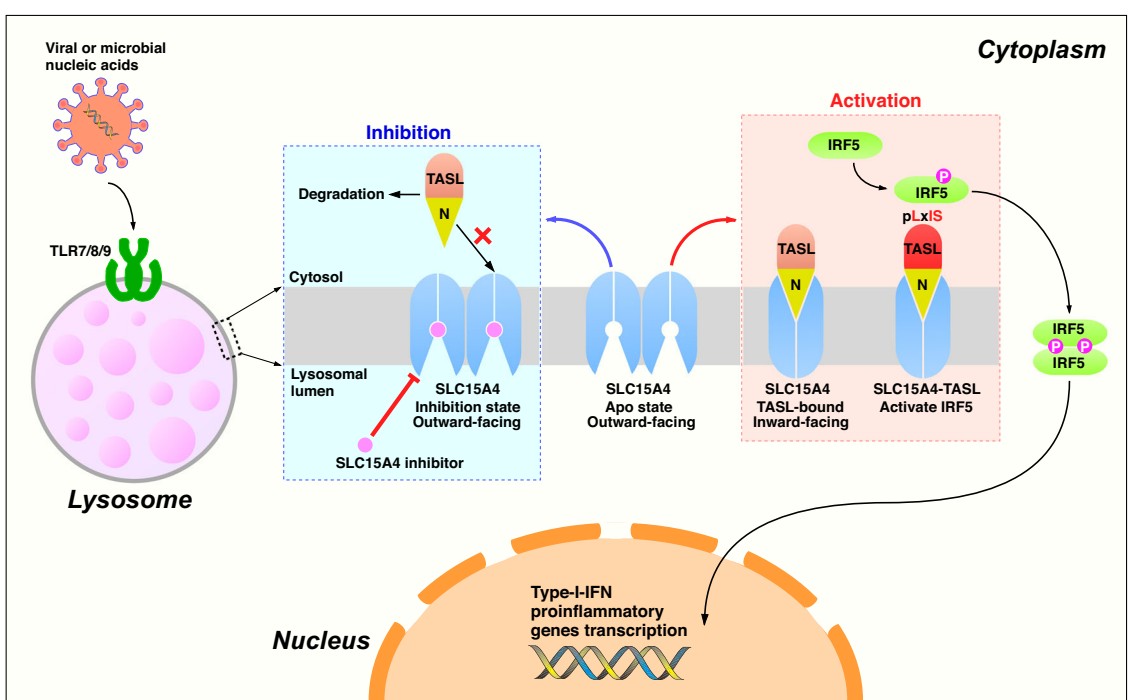

**Fig. 5 | Proposed mechanism of SLC15A4−TASL signaling.** Schematic model of the TASL- and SLC15A4-dependent IRF5 activation in endolysosomal TLR signaling. Apo SLC15A4 is located on the lysosomal lumen and adopts an outward-facing conformation. During the recruitment of TASL, SLC15A4 undergoes an outward-facing to an inward-facing conformational change, leading to the opening of the TASL-binding pocket and the subsequent binding to TASL from the cytosolic side. Activation of TLR7−9 triggers the phosphorylation of the SLC15A4-TASL module, resulting in the following recruitment and activation of transcription factor IRF5. The phosphorylated IRF5 dimer then enters the nucleus to regulate the production of inflammatory cytokines. SLC15A4 can be locked in the outward-facing conformation by a selective inhibitor, which blocks the SLC15A4-TASL interaction, thus interrupting the endolysosomal TLR signaling pathway. The red and blue dashed boxes indicate the activation and inhibition process of SLC15A4, respectively.

signaling. In addition, our findings provide an important framework for structure-guided drug discovery targeting SLC15A4- or TASL-related human diseases.

## Methods
### Materials
The following reagents were purchased from Sigma-Aldrich: NaCl (Cat# S9888), HEPES (4-(2-Hydroxyethyl)piperazine-1-ethane sulfonic acid; Cat# H3375), DTT (1,4-dithiothreitol; Cat# D0632), EDTA (ethylenediaminetetraacetic acid; E9884), Anti-FLAG Affinity Gel (Cat# A2220) and 3×FLAG peptides (Cat# F4799).

In addition, protease Inhibitor Cocktail Tablets (Cat# 04693116001) were obtained from Roche. 10:1 LMNG/CHS premixed solution (Cat# NG310-CH210) were obtained from Anatrace. Strep-Tactin resin (Cat# 2-1208-500) and D-desthiobiotin (Cat# 2-1000-005) were ordered from IBA. We obtained the Polyethylenimine (PEI, Linear, MW 25000; Cat# 23966) from Polysciences. The VigoFect (Cat# T001) was purchased from Vigorous Biotechnology. Hoechst33342 (Cat# H1399) was obtained from Thermo Fisher Scientific. For molecular cloning, the Seamless Assembly Cloning Kit (Cat# C5891) and QuikChange Site-Directed Mutagenesis Kit (Cat# 200518) were purchased from CloneSmarter and Agilent, respectively. For immunoblotting, an anti-Strep antibody (Cat# BE2076, dilution: 1:3000) and anti-Flag antibody (Cat# BE2004, dilution: 1:3000) were ordered from Bioeasytech, human CXorf21 mouse polyclonal antibody, and human SLC15A4 rabbit polyclonal antibody were obtained from Abcam (Cat# ab69152, Dilution: 1:1000) and Abmar (Cat# PHC6279, Dilution: 1:500), respectively. For cell culture, the HEK293T cell line (Cat# CRL-3216) and THP1 cell line (Cat# TIB-202) were obtained from ATCC and authenticated by short tandem repeat profiling. HEK293F cell line (Cat# R79007) was purchased from Thermo Fisher Scientific, the cell culture medium (Cat# SMM 293-TII) was obtained from Sino Biological, and the Penicillin-Streptomycin Solution (Cat# SV30010) was ordered from Hyclone.

### Construct and cell culture
The codon-optimized cDNAs for human full-length SLC15A4 (Uniprot: Q8N697 [https://www.uniprot.org/uniprotkb/Q8N697/entry]), human TASL (Uniprot: Q9HAI6 [https://www.uniprot.org/uniprotkb/Q9HAI6/entry]) and the ALFA nanobody[41] were synthesized and validated by sequencing. The constructs for protein expression and cell-based assays were then built and subcloned into appropriate vectors. All plasmids and primer sequences used in this study are shown in Supplementary Table 1. The *E. coli* strain DH5α was used to generate and amplify plasmids for cell transfection, and cells were cultured in LB medium (Sigma) at 37 °C.

For protein expression, HEK293F cells were cultured in a medium supplemented with 1× Penicillin/Streptomycin (Hyclone) in a Multitron-Pro shaker (Infors, 120 rpm) at 37 °C with 5% CO$_2$. For cell-based assays, HEK293T cells were maintained in DMEM supplemented with 10% (v/v) FBS and Penicillin-Streptomycin Solution (Hyclone). THP1 cells were maintained in RPMI1640 supplemented with 10% (v/v) FBS and Penicillin-Streptomycin Solution (Hyclone).

### Protein expression and purification for cryo-EM analysis
For SLC15A4 expression, the coding sequence for human SLC15A4 including a C-terminus tandem twin Strep-tag was cloned into the pcDNA3.1 vector. For SLC15A4/TASL(1−20)-EGFP complex expression, the coding sequence for human SLC15A4 with C-terminus flag-tag was cloned into the pcDNA3.1 vector, and the coding sequence for TASL (1−20)-EGFP including C-terminus twin Strep-tag were cloned into the pCAG vector. For SLC15A4/ALFA_Nanobody complex expression, the coding sequence for human SLC15A4 with the 253−303 residues replaced by an ALFA tag (SRLEEELRRRLTE) was cloned into the pcDNA3.1 vector, and the coding sequence of ALFA nanobody was cloned into the pcDNA3.1 vector. To produce human SLC15A4 protein,

SLC15A4/TASL complex or SLC15A4/ALFA_Nanobody complex, 1 mg SLC15A4 plasmids, or 0.4 mg SLC15A4 plasmids and 0.6 mg TASL(1−20)-EGFP plasmids mixture, or 0.5 mg SLC15A4-ALFA tag plasmids and 0.5 mg ALFA nanobody plasmids were pre-incubated with 2.5 mg PEI in 50 ml fresh medium for 25 min prior to adding the mixture to one-liter cells when cell density reached $1.5 × 10^6$ per milliliter. The transfected cells were cultured for 40 h before harvesting.

For human full-length SLC15A4, SLC15A4/TASL(1−20)-EGFP complex and SLC15A4/ALFA_Nanobody complex purification, similar protocol was used. In brief, for one batch of protein purification, about four liters of transfected cells were harvested by centrifugation at $3000 × g$. Then all procedures below are carried out at 4 °C or on ice. Harvested cells were resuspended in lysis buffer containing 25 mM HEPES (pH 7.4), 150 mM NaCl, 2 mM DTT, 1 mM EDTA, and protease inhibitor, and then lysed by a high-pressure homogenizer. After removal of cell debris by centrifugation at $10,000 × g$ for 45 min, cell membrane fraction was pelleted by a $150,000 × g$ ultracentrifugation for 1 h. The membrane fraction was resuspended and solubilized in lysis buffer plus 1% (w/v) LMNG and 0.1% (w/v) CHS for 2 h with gentle rotation. After ultracentrifugation at $150,000 × g$ for 30 min, the supernatant was passed through a column filled with Strep-Tactin Sepharose resin. The resin was washed 50 CV (column volume) with wash buffer containing 25 mM HEPES (pH 7.4), 150 mM NaCl, 2 mM DTT, 0.005 % (w/v) LMNG, and 0.0005% (w/v) CHS. The target protein was eluted with wash buffer plus 10 mM desthiobiotin. The eluted protein was concentrated to a final volume of approximately 100 μl by a 50 kDa cut-off centrifugal filter (Millipore) and further purified by size-exclusion chromatography (SEC) (Superose 6 5/150, GE Healthcare) in SEC buffer containing 25 mM HEPES (pH 7.4), 150 mM NaCl, 0.003 % (w/v) LMNG and 0.0003% (w/v) CHS. The SEC fractions corresponding to SLC15A4, SLC15A4/TASL complex, or SLC15A4/ALFA_Nanobody complex were collected and verified by SDS−PAGE for Cyro-EM sample preparation. The peak fractions were concentrated to 10−15 mg/ml for grid preparation.

### Electron microscopy sample preparation and imaging
The cryo-EM grids for SLC15A4, SLC15A4/TASL(1−20)-EGFP complex, and SLC15A4/ALFA_Nanobody complex were all prepared using Vitrobot Mark IV (FEI) at 8 °C and 100% humidity. 3−4 μL aliquots of samples at a concentration of 10−15 mg/ml were applied onto glow-discharged holey carbon grids (Quantifoil R1.2/1.3). After a waiting time of 5 s, the grids were blotted for 2−5 s and plunged into liquid ethane for quick freezing. The grids were screened on a Tecnai Arctica microscope (FEI) operated at 200 kV using a Falcon 3 direct electron detector (FEI).

For SLC15A4/TASL complex the qualified grids were transferred into a Titan Krios microscope (FEI) operated at 300 kV equipped with an energy filter (slit width 20 eV; GIF Quantum LS, Gatan) for data acquisition. Images were recorded using a K3 submit direct electron detector (Gatan) in a super mode at a nominal magnification of ×105,000, corresponding to a calibrated pixel size of 0.42165 Å. Date acquisition was performed automatically using AutoEMation2.0[46] in a movie mode, with a frame exposure time of 0.04 s and a total exposure time of 1.28 s, resulting in a total of 32 frames per stack and the total dose for each stack was approximately 50 e$^-$ Å$^{-2}$. All 32 frames in each stack were aligned and summed using the whole-image motion correction program MotionCor2[47] and binned to a pixel size of 0.8433.

For SLC15A4 in apo state, the qualified grids were transferred into a Titan Krios microscope (FEI) operated at 300 kV equipped with a Cs image corrector and an energy filter (slit width 20 eV; GIF Quantum LS, Gatan) for data acquisition. Images were recorded using a K3 submit direct electron detector (Gatan) in a super mode at a nominal magnification of ×64,000, corresponding to a calibrated pixel size of 0.54895 Å. Date acquisition was performed automatically using AutoEMation2.0[46] in a movie mode, with a frame exposure time of

0.08 s and a total exposure time of 2.56 s, resulting in a total of 32 frames per stack and the total dose for each stack was approximately 50 e⁻ Å⁻². All 32 frames in each stack were aligned and summed using the whole-image motion correction program MotionCor2[47] and binned to a pixel size of 1.0979.

For SLC15A4/ALFA_Nanobody complex, the qualified grids were transferred into a Titan Krios microscope (FEI) operated at 300 kV equipped with an energy filter (slit width 20 eV; GIF Quantum LS, Gatan) for data acquisition. Images were recorded using a K3 submit direct electron detector (Gatan) in a super mode at a nominal magnification of ×105,000, corresponding to a calibrated pixel size of 0.4187 Å. Date acquisition was performed automatically using AutoEMation2.0[46] in a movie mode, with a frame exposure time of 0.08 s and a total exposure time of 2.56 s, resulting in a total of 32 frames per stack and the total dose for each stack was approximately 50 e⁻ Å⁻². All 32 frames in each stack were aligned and summed using the whole-image motion correction program MotionCor2[47] and binned to a pixel size of 0.8374.

### Image processing and 3D reconstruction

For the dataset of SLC15A4/TASL(1–20)-EGFP complex, SLC15A4 and SLC15A4/ALFA_Nanobody complex and 4788 micrographs (movie stacks), 1960 micrographs and 4965 micrographs were collected, respectively. The image processing steps were carried out using cryoSPARC[48]. For the dataset of SLC15A4/TASL(1–20)-EGFP complex, micrographs were imported and the CTF corrections were performed using Patch CTF estimation. After the deletion of bad micrographs, 200 micrographs were used for automatic picking by blob picker, and these particles were subjected to 2D classification. The class averages representing projections of the SLC15A4/TASL complex in different orientations were chosen as templates for template picking from the whole dataset. A total of 6,471,651 particles were picked from 4713 micrographs. These particles were extracted and binned 4 times and subjected to 2D classification. After three rounds of 2D classification, ~2751 K particles in good 2D averages were chosen. These particles were re-extracted and binned 2 times, then subjected to the ab initio reconstruction and the following heterogeneous refinement. The map of each class was measured in Chimera[49], and the particles from good classes were selected and then subjected to the next round of ab initio reconstruction and heterogeneous refinement to further remove bad particles. After two rounds of this operation, 111,027 particles from the best class were selected and re-extracted to the original pixel size 0.8433 Å, and subjected to another three rounds of ab initio reconstruction and heterogeneous refinement to further improve data quality. Finally, the best 87,230 particles were used for 3D reconstruction by non-uniform (NU) refinement[50] to yield a map at 3.03-Å resolution map with no symmetry imposed. The local resolution map was calculated using local resolution estimation in cryoSPARC and displayed in ChimeraX[51]. Similar data processing procedures were applied for a dataset of the SLC15A4 in the apo state and the SLC15A4/ALFA_Nanobody complex. Finally, a 2.93-Å resolution map for SLC15A4 and a 3.25-Å resolution map for SLC15A4/ALFA_Nanobody complex were obtained. Please refer to Supplementary Figs. 3, 7, 10 for the workflow of image processing.

### Model building

The density maps of SLC15A4/TASL complex, SLC15A4 in apo state, and SLC15A4/ALFA_Nanobody complex were of sufficient quality for de novo model building in COOT[52] (Supplementary Figs. 3, 7, 10), facilitated by the structure of human full-length SLC15A4 predicted on AlphaFold[53,54]. For SLC15A4/TASL complex model building, the predicted model of human SLC15A4 was docked into the cryo-EM map with a resolution of 3.03 Å in Chimera and manually adjusted in COOT to acquire the atomic model of SLC15A4 and the bound TASL[49,55]. Model refinement was performed on the main chain of the

two atomic models using the real_space_refine module of PHENIX[56] with secondary structure and geometry restraints to avoid overfitting. After manual adjustment in COOT, the models were subjected to real-space refinement in PHENIX. For SLC15A4 in the apo state and the SLC15A4/ALFA_Nanobody complex model building, similar procedures were used. Cryo-EM data collection and refinement statistics are shown in Supplementary Table 2.

### Constructs and cellular fluorescence co-localization

SLC15A4 fused to mCherry or Venus were cloned into pcDNA3.1 vector, TASL or TASL truncates fused to mEGFP were cloned into pCAG vector, LAMP1 fused to mCherry was cloned into PLVX backbone vector. For low expression of TASL-mEGFP, the CAG promoter was replaced by the UBC promoter. All the fluorescent expression vectors contain a 3×GGGS flexible linker between interest protein and fluorescent protein. Different pairs of plasmids for fluorescence co-localization are transfected into HEK293T cells with vigofect. Cells were maintained in DMEM supplemented with 10% (v/v) FBS and Penicillin-Streptomycin Solution (Hyclone). 48 h after transfection, cells were stained with Hoechst33342 and then washed with PBS. Fluorescence Images were acquired on a confocal microscopy (Nikon A1 HD25).

### Analyses of the SLC15A4-TASL interaction by immunoprecipitation

To determine the formation of SLC15A4-TASL complexes in cells, SLC15A4-FLAG wild-type or mutants were co-transfected with TASL(1–20)Strep into HEK293T cells. Whole-cell lysates were prepared in lysis buffer (25 mM HEPES, 300 mM NaCl, 5 mM EDTA, 1% NP-40, pH 7.5) and cleared by centrifugation at 13,000 × g, 10 min, 4 °C. Take out 100 μl whole-cell lysate as input, and the rest was incubated with FLAG beads overnight at 4 °C. Beads were washed at least five times with the lysis buffer and then detect the co-precipitated proteins by immunoblotting.

### Mass spectrometry analysis for SLC15A4 glycosylation identification

For sample preparation, freshly purified SLC15A4 proteins were subjected to SDS–PAGE, then the gel bands of SLC15A4 were excised and reduced with 5 mM of DTT and alkylated with 11 mM iodoacetamide which was followed by in-gel digestion with sequencing grade modified trypsin (Promega, Madison, WI) in 50 mM ammonium bicarbonate at 37 °C overnight. The sample was quenched by adding 10% (v/v) trifluoroacetic acid (TFA) to adjust the pH to below 2. The peptides were extracted twice with 0.1% TFA in a 50% acetonitrile aqueous solution for 1 h and then dried in a speedVac. Peptides were dissolved in 25 μl 0.1% TFA, and 6 μl of the extracted peptides was then subjected to liquid chromatography-tandem mass spectrometry (LC-MS/MS) analysis.

For LC-MS/MS, the peptides were separated by a 60 min gradient elution at a flow rate 0.30 μl/min with a Dionex Ultimate 3000 HPLC system (Thermo Fisher Scientific), which was directly interfaced with an Orbitrap Fusion mass spectrometer (Thermo Fisher Scientific). The analytical column was a homemade fused silica capillary column (75 μm ID, 150 mm length) packed with C-18 resin (100 Å, 2 μm, Dr.Maisch). Mobile phase A consisted of 0.1% formic acid, and mobile phase B consisted of 100% acetonitrile and 0.1% formic acid. An LTQ-Orbitrap mass spectrometer was operated in the data-dependent acquisition mode using Xcalibur 4.3.73.11 software and there was a single full-scan mass spectrum in the Orbitrap (300–1500 *m/z*, 120,000 resolution) followed by 3 s data-dependent MS/MS scans in an Ion Routing Multipole at stepped 27, 30, 33 normalized collision energy (HCD).

For data analysis, glycopeptide fragmentation data were extracted from the raw file using Byonic (Version 2.8.2). The MS data was searched using the Protein Metrics 309 N-glycan library. The search

criteria were as follows: Non-specificity; carbamidomethylation (C) was set as the fixed modifications; the oxidation (M) was set as the variable modification; precursor ion mass tolerances were set at 20 ppm for all MS acquired in an orbitrap mass analyzer; and the fragment ion mass tolerances were set at 0.02 Da for all MS2 spectra acquired. The intensities of the same glycopeptide in each site were combined and analyzed for proportion. Data with a score under 30 and abnormal values with intensity above $1 \times 10^{10}$ were removed. The glycans were classified into oligomannose, hybrid, complex, and core type based on composition. Hybrid and complex type glycan were subdivided according to fucose component and antenna. The mean of the two replicates represents the ratio of each glycan type.

## Mass spectrometry for SLC15A4-bound cholesterol identification

For sample preparation, freshly purified SLC15A4 proteins were subjected to ultra-performance liquid chromatography (UPLC)-MS/MS analysis.

For UPLC-MS/MS, the UPLC system was coupled to a Q-Exactive HFX orbitrap mass spectrometer (Thermo Fisher Scientific) equipped with a heated electrospray ionization (HESI) probe. 2 μl protein sample was loaded to a Kinetex® Biphenyl column (2.1 × 150 mm, 2.6 μm, Phenomenex) for the positive mode. Separation was initiated at 80% mobile phase B with a flow rate of 300 μl/min, then the sample was eluted to an orbitrap mass spectrometer with acetonitrile containing 0.1% formic acid as eluent from 80% to 99% within 6.5 min. Untargeted metabolites screening is performed on Q-Exactive HFX Orbitrap mass spectrometer (Thermo Fisher Scientific) after calibrating following manufacturer's guidelines. Data with mass ranges of m/z 300–500 at positive ion mode with data-dependent MS/MS acquisition. The full-scan and fragment spectra were collected with resolutions of 60,000 and 15,000, respectively. The source parameters are as follows: spray voltage: 3,200 V; capillary temperature: 320 °C; heater temperature: 300 °C; sheath gas flow rate: 35 Arb; auxiliary gas flow rate: 10 Arb.

Data analysis was performed by the software Xcalibur 4.4 (Thermo Fisher Scientific) Cholesterol assignment was confirmed using chemical standards based on retention time.

## FRET measurement

SLC15A4-GFP was co-transfected with SLC15A4-mCherry into HEK293T cells. At the same time, GFP and mCherry were co-transfected into HEK293T cells as a negative control, and the TASL(1–20)-GFP and SLC15A4-mCherry were co-transfected into HEK293T cells as a positive control. Forty-eight hours after transfection, a Glass bottom plate with transfected cells was mounted onto a Nikon A1 HD25 microscope, and the dequenching FRET images were captured at the same laser bleaching power. GFP was excited with 488 nm laser line and visualized using the 515/30 bandpass filter, and mCherry was excited with the 561 nm laser line, visualized using the 595/30 bandpass filter. All the images were processed with NIS-Elements. The FRET efficiency was calculated with the formula FRET efficiency = (DQ-Q)/DQ, DQ, and Q represented dequenched and quenched donor fluorescence intensity, respectively. The schematics of FRET efficiency were generated using GraphPad Prism 8.2.1.

## Genes knock out and rescue experiments

The genes KO THP1 cell lines and genes stable expression for rescue cell lines are constructed through Lentiviral gene transduction. CRISPR–Cas9-based knockout cell line generation was performed using pLentiCRISPRv2 with the reported sgRNAs[11] (sgRenilla: GGTATAATACACCGCGCTAC, sgTASL: GTAGAAATGGAATCCTCCAT, sgSLC15A4: GGAGCGATCCTGTCGTTAGG). For Rescue experiments, Codon-optimized cDNA (WT or mutant) was cloned into a pLvx-based backbone of which the CMV promoter is replaced with the UBC promoter. HEK293T cells were transfected with the respective lentiviral

vectors and packaging plasmids using vigofect or PEI. The medium was exchanged to RPMI, supplemented with 10% (v/v) FBS and antibiotics 24 h later, and the cell supernatants were collected for infection after 48 h transfection. THP1 cells were infected using the collected supernatants (with spin infection 2000 × g 45 min for lines used in Supplementary Fig. 6) and were selected with the respective antibiotics 48 h after infection. Selected cell populations were used for experimental investigations without further subcloning to avoid clonal effects.

## R848 treatment and enzyme-linked immunosorbent assay

For IL-8 analysis, THP1 cells with cell density $1.0 \times 10^6$ per milliliter were plated on a 24-well plate and treated with R848 (2 μg/ml). After 24 h, the cell supernatant was collected for enzyme-linked immunosorbent assay (ELISA). The human IL-8 ELISA experiments were carried out according to the manufacturer's instructions. ELISA kits for human IL-8 (Cat# KIT10098) were from Sino Biological.

For CCL2 and TNF analysis, THP1 cells ($1 \times 10^5$/200 μl) were plated on 96-well plates and treated with R848 (5 μg/ml; Cat# tlrl-r848; InvivoGen) for 24 h. Alternatively, $2 \times 10^5$ cells were plated in 120 μl and stimulated with TL-8 (0.5 μg/ml) for IFNβ production. The cell supernatant was collected after 24 h and analyzed with the human CCL2 (Cat# 88-7399-88) and TNFα (Cat# 88-7346-88) ELISA kits from Invitrogen, or the IFNβ (Cat# DY814-05) DuoSet kit from R&D, following manufacturer's instructions.

## R848 treatment and *IFNB* quantitative real-time PCR

THP1 cells ($1 \times 10^6$/500 μl) were stimulated with R848 (5 μg/ml) for *IFNB* transcription induction. After 0-3 h, the cells were collected, total RNA was extracted using the ReliaPrep RNA cell miniprep system (Cat# Z6011, Promega) and RNA was retrotranscribed to cDNA using the RevertAid First Strand cDNA Synthesis kit through oligo-(dT) primers (Cat# K1622, ThermoFisher). Finally, *IFNB* transcription was quantified in a quantitative real-time PCR (qPCR) with the KAPA SYBR FAST kit (Cat# KK4611, Kapa Biosystems) in a Lightcycler 480 (Roche) as in ref. 57. We used the human *IFNB* forward TGCTCTCCTGTT GTGCTTCTCCAC and reverse primer ATAGATGGTCAATGCGGC GTCC, and the human *HPRT1* as control, with the forward AGACTTTGCTTTCCTTGGTCAG and reverse primer CCAACAAA GTCTGGCTTATATCC. Ct values were calculated using the $2^{nd}$ Derivative method in the LightCycler Software version 1.5 (Roche), and gene expression was determined using the $2^{-\Delta Ct}$ method, with *HPRT1* as reference.

## Western blot analysis

Cells ($1–2 \times 10^6$) were lysed in RIPA lysis buffer (25 mM Tris, 150 mM NaCl, 0.5% NP-40, 0.5% deoxycholate (w/v), 0.1% SDS (w/v), pH 7.4) supplemented with complete EDTA-free protease inhibitor cocktail (Cat# 11836170001, Roche), Halt phosphatase inhibitor cocktail (Cat# 1862495, Thermo Fisher Scientific) and Benzonase (Cat# 71205, Merck). Lysates were cleared by centrifugation (13,000 × g, 10 min, 4 °C), and protein was quantified with BCA (Cat# 23225, Thermo Fisher Scientific). Cell lysates for SLC15A4 blotting were incubated with PNGase F (100 U for 50 μl of lysate, Cat# P0704L, NEB) for 30 min at 37 °C. Then, 10–15 μg of protein lysate were resolved by regular or Phos-tag Acrylamide (200 μM, Cat# 304-93521, FujiFilm) SDS–PAGE with PageRuler Plus prestained protein ladder as reference (Cat# 26616, Thermo Fisher Scientific), and blotted to nitrocellulose membranes (Amersham) at 100 V, 0.3 A for 1 h. Prior to transfer, Phos-tag gels were incubated with transfer buffer containing 10 mM EDTA (Sigma) and then washed in transfer buffer. After transfer, membranes were blocked with 5% non-fat dry milk in TBS-T and probed with antibodies: custom-made rabbit anti-SLC15A4 (Genscript, dilution: 1:1000, described in ref. 11), custom-made rabbit anti-TASL (Eurogentec, dilution: 1:1000)[57], rabbit anti-SAPK/JNK (Cat# 9252, dilution: 1:1000), rabbit anti-phospho-SAPK/JNK (Cat# 4668, dilution: 1:1000),

mouse anti-IkBα (Cat# 4814, dilution: 1:1000), rabbit anti-phospho-IkBα (Cat# 2859, dilution: 1:1000) (all from Cell Signaling), mouse anti-Flag M2 (Cat# F1804, dilution: 1:1000), rabbit anti-TASL (CXorf21; Cat# HPA001185, dilution: 1:1000) (both from Sigma), rabbit anti-IRF5 (Cat# ab181553, Abcam, dilution: 1:1000), mouse anti-GAPDH (Cat# sc-365062, Santa Cruz, dilution: 1:1000), and the secondary antibodies goat anti-rabbit IgG (H + L) HRP (Cat# 111-035-003, dilution: 1:1000), goat anti-mouse IgG (H + L) HRP (Cat# 115-035-003, dilution: 1:1000) (both from Jackson Immuno Research). In experiments in which multiple antibodies were used, equal amounts of samples were loaded on multiple SDS–PAGE gels and western blots sequentially probed with a maximum of three antibodies.

### Statistics & reproducibility
Data are represented as individual values, mean, mean ± s.d. as described in the figure legends. Group sizes ($n$) are indicated in the figure or figure legends. No statistical methods were used to pre-determine the sample size. Data exclusions and Randomization were used only in cryo-EM data processing and structure refinement in this study. The low-quality data such as bad micrographs or particles were excluded to reach high resolution, and the particles were randomly assigned to two independent groups in refinement. The exclusion and randomization algorithms are implemented in cryoSPARC. The researchers were not blinded to allocation during experiments and outcome assessment.

### Reporting summary
Further information on research design is available in the Nature Portfolio Reporting Summary linked to this article.

## Data availability
The 3D cryo-electron microscopy density map and the coordinates of atomic models has been deposited in the Electron Microscopy Data Bank (EMDB) and the Protein Data Bank (PDB) with the following accession codes: EMD-36753 and 8JZU for SLC15A4-TASL complex; EMD-36752 and 8JZS for SLC15A4 apo dimer state; EMD-36751 and 8JZR for SLC15A4 apo monomer state. The other structures used for comparative analysis in this study can be found in the PDB with the following accession codes: 7PMW, 7PMX and 7PN1 for SLC15A1; 7NQK and 7PMY for SLC15A2. All protein sequences used in this study are available at Uniprot (https://www.uniprot.org/) with the following accession codes: Q8N697 for human SLC15A4; Q9HAI6 for human TASL; P46059 for Human SLC15A1; Q16348 for Human SLC15A2; Q8IY34 for Human SLC15A3; Q91W98 for mouse SLC15A4; O09014 for rat SLC15A4; A6QQL0 for bovin SLC15A4; Q68F72 for African clawed fog SLC15A4; Q9D3J9 for mouse TASL; Q32LD7 for bovin TASL; F1SPV1 for pig TASL; G1U138 for rabbit TASL. Source data are provided as a Source Data file. Source data are provided with this paper.

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

## Acknowledgements

We thank Dr. Xiaomin Li and Dr. Fan Yang (Tsinghua University) for technical support during EM image acquisition. We thank the Cryo-EM Facility Center of Tsinghua University Branch of China National Center for Protein Sciences (Beijing) for providing the facility support. We also thank Dr. Chuangye Yan (Tsinghua University) and Dr. Peilong Lu (Westlake University) for their constructive suggestions on construct design and protein purification. We would like to acknowledge the assistance of SLSTU-Nikon Biological Imaging Center of Tsinghua University for technical support during Confocal microscopy image acquisition. The computation was completed on the Yang laboratory GPU workstation. This work was supported by funds for M.Y. from the National Key R&D Program of China (2022YFA1302701), the National Natural Science Foundation of China (32030056), the Tsinghua-Foshan Innovation Special Fund (TFISF-2022THFS6122) and the King Abdullah University of Science and Technology (KAUST) Office of Sponsored Research (OSR) under Award No. OSR-2020-CRG9-4352; the grants for X.C from the Postdoctoral Fellowship of Tsinghua-Peking Center for Life Sciences, and the General Financial Grant from the China Postdoctoral Science Foundation (2017M62075). The work in M.R. laboratory is supported by the Swiss National Science Foundation (Project grant 310030_200709) and the Fondation Pierre Mercier pour la science. G.S.-F. is supported by funding from the Austrian Academy of Sciences.

## Author contributions

M.Y. conceived and supervised the study. M.X. and X.C. performed the construct design, protein purification and detergent screening with the help of J.Y., Y.Z., Z.Y., Q.C. and B.Z.; X.C. and S.Z. performed EM sample preparation, data collection and structural determination with the help of C.L. and Y.W.; M.X. performed most of the cell-based study with the help of M.S., Q.D., J.N. and L.D.; M.M.T., M.D. and M.R. performed, analyzed and drew the figures on the cell-based study presented in Supplementary Fig. 6; X.T. and H.D. carried out mass spectrometry analysis for glycoslation identification; L.X. and X.L. carried out mass spectrometry analysis for cholesterol identification; X.C., M.X. and S.Z. built the model, drew the figures, and wrote the manuscript with the help of W.Z., G.S.-F., M.R. and M.Y; All authors contributed to discussion of the data and editing the manuscript.

## Competing interests

CeMM and the Medical University of Vienna are the applicants of European priority patent applications (EP 22 203 423.3, EP 22 203 422.5, EP 22 203 421.7, status: filed) in which M.R. and G.S.-F are listed as

inventors, covering small-molecule modulators of TASL and their medical use. G.S.-F. is founder and shareholder of Solgate, a company focused on SLC inhibitors. The other authors declare no competing interests.

## Additional information

[1]Ministry of Education Key Laboratory of Protein Science, Tsinghua-Peking Center for Life Sciences, Beijing Advanced Innovation Center for Structural Biology, School of Life Sciences, Tsinghua University, 100084 Beijing, China. [2]School of Medicine, Tsinghua University, 100084 Beijing, China. [3]Department of Immunobiology, University of Lausanne, 1066 Epalinges, Switzerland. [4]Beijing Life Science Academy, 102209 Beijing, China. [5]Department of Nephrology, First Medical Center of Chinese PLA General Hospital, Nephrology Institute of the Chinese People's Liberation Army, National Key Laboratory of Kidney Diseases, National Clinical Research Center for Kidney Diseases, Beijing Key Laboratory of Kidney Disease Research, 100853 Beijing, China. [6]Center for Infectious Disease Research, School of Medicine, Tsinghua University, 100084 Beijing, China. [7]Department of Cardiology, Beijing Tsinghua Changgung Hospital, School of Clinical Medicine, Tsinghua University, 102218 Beijing, China. [8]MOE Key laboratory of Bioinformatics, School of Life Sciences, Tsinghua University, 100084 Beijing, China. [9]Metabolomics and Lipidomics Center at Tsinghua—National Protein Science Facility, School of Life Sciences, Tsinghua University, 100084 Beijing, China. [10]Beijing Advanced Innovation Center for Structural Biology, Technology for Protein Research, School of Life Sciences, Tsinghua University, 100084 Beijing, China. [11]CeMM Research Center for Molecular Medicine of the Austrian Academy of Sciences, Vienna, Austria. [12]Center for Physiology and Pharmacology, Medical University of Vienna, Vienna, Austria. [13]Cryo-EM Facility Center, Southern University of Science & Technology, 518055 Shenzhen, Guangdong, China. [14]These authors contributed equally: Xudong Chen, Min Xie, Sensen Zhang. ✉e-mail: maojunyang@tsinghua.edu.cn

