## [Peer Review File · Nature Communications]

Reviewers' Comments:

Reviewer #1:

Remarks to the Author:

This manuscript by Chen et al reports cryo-EM structural characterization of the interaction between SLC15A4 and TASL, which plays an essential role in mediating endosomal Toll-like receptor signaling. TASL was recently identified as an innate immunity adaptor that is required for IRF5 activation downstream of Toll-like receptors. This process depends on the interaction of TASL with SLC15A4, a lysosomal solute carrier protein. This study reveals the details of this interaction, involving the N-terminal end of TASL and a cytosolic side pocket of SLC15A4 in the inward-facing conformation, after overcoming a number of technique challenges through different tricks. A somewhat unexpected dimeric state of SLC15A4 is also presented. The structures presented in the manuscript appear to be of high quality, and provide a clear model for how the conformational states of SLC15A4 is tightly coupled to the interaction with TASL. The associated biochemical and functional assays validate the structural model. This is an important study regarding the understanding of the mechanism of TASL, and will be useful for drug development targeting the SLC15A4/TASL interaction. I only have some minor suggestions, mostly directed at making the presentation clearer for readers:

1. Line 84, the "pLxIS" motif. It may not be obvious to some readers that this represents a phosphorylated motif. It would be better to say something like "a LxIS motif that can be phosphorylated by the upstream kinase TBK1". Related to this, please define terms such as "SLE" and "RA" at the first occurrence.
2. Lines 129-131, presumably the strep-tag here are attached to the C-terminus of the protein, because the N-terminal is required for binding SLC15A4. It would be helpful to mention this point as the location of the tag actually matters in this case.
3. Lines 135-140, the sentence is long and confusing.
4. Line 141, it is more appropriate to call it luminal domain instead of extracellular domain.
5. Line 145-146, "spatial volume" could simply be "volume", and "stoichiometric ratio" could be just "stoichiometry".
6. Line 150, again, this domain is not extracellular, it is in the lysosomal lumen. It should be called luminal domain.
7. Line 153, N- and C-termini.
8. SLC15A4 is labelled S-mcherry in Figure 2a and b, while it is labelled S154 in Figure 3d-g. It is better to make the labels consistent to avoid confusion.
9. The results shown in Figure 3d, f and g lack negative controls. Negative controls with dimer disrupting mutants should be provided. The observation of the dimer is an interesting point, and its existence seems convincing from the structural point of view. The data without proper negative control do not add much to this point. If adding the negative control would take too much time, it seems ok to leave these data out to avoid unnecessary delay of the publication.
10. Line 285. There is no such thing as "soluble signal", I guess that the authors meant signal from the nanobody bound to the soluble ALFA tag.
11. Line 360. Typo, "recluted" should be "recruited".
12. The FSC curves in the PDB validation reports do not make sense, as the blue unmasked-calculated FSC remaining 1 throughout the spatial frequency ranges. Did the authors upload the two independent half maps?

Reviewer #2:

Remarks to the Author:

In this study, the authors report the structure of SLC15A4-TASL complex, which is an important mediator of Toll signaling from endosomes. The structural analysis is well done, with 3 angstrom resolution reported. Specific amino acids in both proteins were identified that control protein complex formation and signal transduction in THP1 cells. It is this latter point that I would like to have more clarity on.

In a revised study, the authors are encouraged to examine more TLR-dependent activities that IL-8 expression (Figure 2E, F). This minimal functional analysis is of questionable significance, as Type I Interferons are the key factors induced by TLRs via SLC15A4-TASL. Additionally, TBK1 and IRF5 phosphorylation assays should be performed.

This additional functional analysis will enhance the quality of this study.

Reviewer #3:

Remarks to the Author:

Three crucial conformational structures of SLC15A4 are solved by cryo-EM: the apo monomeric and dimeric form (with 4 cholesterol molecules at the interface); and the monomeric TASL (N terminal fragment) bound form. Some conformational changes accompanying the switch from an outward facing to inward facing state have been noted. The work is important as it reveals possible mechanisms of regulatory switches that might become useful in future drug discovery of human autoimmune diseases involving SLC15A4-TASL interactions.

The work is highly relevant and performed according to high standards: whenever possible, the supporting controls have been carried out. For example, generation of mutants to confirm the relevance of the identified TASL-SLC15A4 interacting amino acids and co-immunoprecipitations. In addition, the size or flexibility of the monomeric SLC15A4 prevented its structure determination by cryo-EM. One way to solve this, would be to generate antibodies or nanobodies against the SLC15A4 molecule, however, this apparently failed as well. This impediment was solved by replacing an external loop of the SLC15A4 between transmembrane helix 6 and 7 by the ALFA tag (i.e. a short strong α helical forming peptide) and using the cognate nanobody against this helical peptide tag. This is a novel application of the underused and under appreciated alpha tag and its dedicated nanobodies.

The results are well explained and (relatively) easy to follow, despite it is a confusing and complex pathway that is used by Toll like receptors to recognise pathogens and inducing the release of host protecting interferons and cytokines of the innate immune response.

The cited work in the bibliographic list is appropriate.

Reviewer #4:

Remarks to the Author:

In this work, the authors used cryo-EM and cell-based assays to investigate the mechanisms of TASL-SLC15A4 interaction. In general, the in vitro experimental setups seem appropriate and the relevant data seem reliable, but those that are pertaining to cellular imaging assays raise some concerns. Specifically, the authors claimed that BIFC could confirm that SLC15A4 exists as dimers in cells. However, the split venus fragments used in this study are prone to nonspecific and irreversible self-complementation, which generate nonspecific signals independent of target protein interactions. Therefore, it is very likely that the observed signal in Fig. 3E stems from undesirable split venus fragment complementation to some extent, rather than being entirely reflecting SLC15A4 dimerization as claimed. Similarly, the ensemble FRET assay used in the study cannot provide quantitative information regarding both the extent of oligomerization and the conformation of a protein. It is possible that a group of randomly oriented SLC15A4-EGFP and SLC15A4-mCherry molecules being in very close proximity to one another (but not associating)

can produce similar FRET signals.

Minor comments

- 1) One Page 11, line 256 "homotyopic" should be "homotypic", and line 261 "pla" is a typo.
- 2) One Page 10, line 256 of Extended data. "efficiency" should be "efficiency".

REVIEWER COMMENTS

Reviewer #1 (Remarks to the Author):

This manuscript by Chen et al reports cryo-EM structural characterization of the interaction between SLC15A4 and TASL, which plays an essential role in mediating endosomal Toll-like receptor signaling. TASL was recently identified as an innate immunity adaptor that is required for IRF5 activation downstream of Toll-like receptors. This process depends on the interaction of TASL with SLC15A4, a lysosomal solute carrier protein. This study reveals the details of this interaction, involving the N-terminal end of TASL and a cytosolic side pocket of SLC15A4 in the inward-facing conformation, after overcoming a number of technique challenges through different tricks. A somewhat unexpected dimeric state of SLC15A4 is also presented. The structures presented in the manuscript appear to be of high quality, and provide a clear model for how the conformational states of SLC15A4 is tightly coupled to the interaction with TASL. The associated biochemical and functional assays validate the structural model. This is an important study regarding the understanding of the mechanism of TASL, and will be useful for drug development targeting the SLC15A4/TASL interaction. I only have some minor suggestions, mostly directed at making the presentation clearer for readers:

1. Line 84, the “pLxIS” motif. It may not be obvious to some readers that this represents a phosphorylated motif. It would be better to say something like “a LxIS motif that can be phosphorylated by the upstream kinase TBK1”. Related to this, please define terms such as “SLE” and “RA” at the first occurrence.

Response: We fully agree with the point made by the Referee and have added the definitions of these terms in the revised manuscript. (line 97-98, 104-105). Concerning the kinase responsible for TASL phosphorylation, this is still unknown. The most likely candidate is IKK beta, while TBK1 seems not to be required (see ref 5 – Heinz et al).

2. Lines 129-131, presumably the strep-tag here are attached to the C-terminus of the protein, because the N-terminal is required for binding SLC15A4. It would be helpful to mention this point as the location of the tag actually matters in this case.

Response: We thank the Referee for the kind reminder. We have included an explanation regarding the location of the protein tag in the revised manuscript (line 143-144, 150-151).

3. Lines 135-140, the sentence is long and confusing.

Response: We have rewritten this sentence in the revised manuscript (line 148-153).

4. Line 141, it is more appropriate to call it luminal domain instead of extracellular domain.

Response: We thank the Referee for the correction. We have corrected this mistake in the revised manuscript by removing this comment as in cells TASL faces the cytoplasmic side (line 156).

5. Line 145-146, “spatial volume” could simply be “volume”, and “stoichiometric ratio” could be just “stoichiometry”.

Response: We thank the valuable suggestion from the Referee. We have corrected them in the revised manuscript (line 160-161).

6. Line 150, again, this domain is not extracellular, it is in the lysosomal lumen. It should be called luminal domain.

Response: Thank again for the careful review of the Referee. We have corrected this mistake in the revised manuscript (line 165).

7. Line 153, N- and C-termini.

Response: We apologize for this mistake, and we have corrected this error (line 167-168).

8. SLC15A4 is labelled S-mcherry in Figure 2a and b, while it is labelled S154 in Figure 3d-g. It is better to make the labels consistent to avoid confusion.

Response: We thank the helpful suggestion from the Referee. In the revised manuscript, SLC15A4 is uniformly labelled as S154 in all figures to make the labels consistent.

9. The results shown in Figure 3d, f and g lack negative controls. Negative controls with dimer disrupting mutants should be provided. The observation of the dimer is an interesting point, and its existence seems convincing from the structural point of view. The data without proper negative control do not add much to this point. If adding the negative control would take too much time, it seems ok to leave these data out to avoid unnecessary delay of the publication.

Response: We appreciate the helpful suggestion made by the Referee. In the revised manuscript, we have removed the Co-IP data and the Bimolecular fluorescence complementation (BIFC) data, and provided new fluorescence resonance energy transfer (FRET) data with positive and negative control as a new figure (Extended Data Fig. 9).

Extended Data Fig. 9

10. Line 285. There is no such thing as “soluble signal”, I guess that the authors meant signal from the nanobody bound to the soluble ALFA tag.

Response: We have rewritten this sentence to avoid confusion (line 304).

11. Line 360. Typo, “recluted” should be “recruited”.

Response: We are very sorry for this obvious mistake, and we have corrected the spelling error in the revised manuscript (line 380).

12. The FSC curves in the PDB validation reports do not make sense, as the blue unmasked-calculated FSC remaining 1 throughout the spatial frequency ranges. Did the authors upload the two independent half maps?

Response: We thank for the concerning about this issue. We have revised and re-upload the correct PDB validation reports.

Reviewer #2 (Remarks to the Author):

In this study, the authors report the structure of SLC15A4-TASL complex, which is an important mediator of Toll signaling from endosomes. The structural analysis is well done, with 3 angstrom resolution reported. Specific amino acids in both proteins were identified that control protein complex formation and signal transduction in THP1 cells. It is this latter point that I would like to have more clarity on.

In a revised study, the authors are encouraged to examine more TLR-dependent activities that IL-8 expression (Figure 2E, F). This minimal functional analysis is of questionable significance, as Type I Interferons are the key factors induced by TLRs via SLC15A4-TASL. Additionally, TBK1 and IRF5 phosphorylations assays should be performed.

This additional functional analysis will enhance the quality of this study.

Response: We appreciate the reviewer's comments for helping us improve the quality of our work. More functional analysis was performed according to the reviewer's suggestions. We investigated the effects of SLC15A4 or TASL mutants on TNF, CCL2 and IFN β induction in THP1 cells. In addition, we monitored activation of IRF5, NF- κ B and MAPK signaling pathways upon TLR7/8 stimulation. These data were obtained using new, independently established THP1 lines generated in the Rebsamen lab, and largely confirmed the effect of the selected TASL and SLC15A4 mutations (New Extended Data Figure 6).

Regarding TASL, E4K and Y6A mutants showed reduced TNF and CCL2 production as well as *IFNB* mRNA induction, while S3A had no or minor effects (Extended Data Fig. 6a, b and c).

Concerning SLC15A4, SLC15A4 E465K and R48A mutants showed full or partial impairment in TNF, CCL2 and IFN β production (Extended Data Fig. 6d, e and f), in line with the effect of these substitutions on TASL binding (Fig. 2b and 2d). The K384A mutant behaved like the wildtype construct in these settings (see Reviewer Figure 1 here below), most likely because the newly established lines expressed the SLC15A4 constructs at higher levels compared to the endogenous SLC15A4 in sgRenilla control cells (Extended Data Fig. 6D). This higher protein expression likely compensates for the minor defect in TASL binding observed for this mutant (Fig. 2b and 2d), as also indicated by the restored level of endogenous TASL observed in this cell line (Reviewer Figure 1d' here below). As we cannot conclude on the effect of K384A in these settings, we did not include this mutant in the Extended Data Figure 6 for clarity.

Lastly, we assessed R848-induced signaling. TASL E4K and Y6A as well as SLC15A4 E465K and R48A mutants displayed impaired or reduced IRF5 activation, which largely correlated with magnitude of the observed reduction in cytokine and IFN β induction. As expected, no difference was observed in NF- κ B and MAPK pathway induction, monitored by phosphorylation of I κ B α and JNK, respectively. Concerning TBK1, our previous data do not support a key role for TLR7/8-induced IRF5 activation, which seem to rely mostly on IKK beta (please refer to our previous publication ref 5 - Heinz et al. Nature 2020).

Extended Data Fig. 6

Reviewer Figure 1 (panel labels related to Extended Data Fig. 6)

Reviewer #3 (Remarks to the Author):

Three crucial conformational structures of SLC15A4 are solved by cryo-EM: the apo monomeric and dimeric form (with 4 cholesterol molecules the interface); and the monomeric TASL(N terminal fragment) bound form. Some conformational changes accompanying the switch from an outward facing to inward facing state have been noted. The work is important as it reveals possible mechanisms of regulatory switches that might become useful in future drug discovery of human autoimmune diseases involving SCL15A4-TASL interactions.

The work is highly relevant and performed according to high standards: whenever possible, the supporting controls have been carried out. For example, generation of mutants to confirm the relevance of the identified TASL-SLC15A4 interacting amino acids and co-immunoprecipitations.

In addition, the size or flexibility of the monomeric SLC15A4 prevented its structure determination by cryo-EM. One way to solve this, would be to generate antibodies or nanobodies against the SLC15A molecule, however, this apparently failed as well. This impediment was solved by replacing an external loop of the SLC15A4 between transmembrane helix 6 and 7 by the ALFA tag (i.e. a short strong α helical forming peptide) and using the cognate nanobody against this helical peptide tag. This is a novel application of the underused and under appreciated alpha tag and its dedicated nanobodies.

The results are well explained and (relatively) easy to follow, despite it is a confusing and complex pathway that is used by Toll like receptors to recognise pathogens and inducing the release of host protecting interferons and cytokines of the innate immune response.

The cited work in the bibliographic list is appropriate.

Response: We are very grateful for the careful review of the Referee.

Reviewer #4 (Remarks to the Author):

In this work, the authors used cryo-EM and cell-based assays to investigate the mechanisms of TASL-SLC15A4 interaction. In general, the in vitro experimental setups seem appropriate and the relevant data seem reliable, but those that are pertaining to cellular imaging assays raise some concerns. Specifically, the authors claimed that BIFC could confirm that SLC15A4 exists as dimers in cells. However, the split venus fragments used in this study are prone to nonspecific and irreversible self-complementation, which generate nonspecific signals independent of target protein

interactions. Therefore, it is very likely that the observed signal in Fig. 3E stems from undesirable split venus fragment complementation to some extent, rather than being entirely reflecting SLC15A4 dimerization as claimed. Similarly, the ensemble FRET assay used in the study cannot provide quantitative information regarding both the extent of oligomerization and the conformation of a protein. It is possible that a group of randomly oriented SLC15A4-EGFP and SLC15A4-mCherry molecules being in very close proximity to one another (but not associating) can produce similar FRET signals.

Response: We appreciate the reviewer's comments for helping us improve the quality of our work. In the revised manuscript, the BIFC data were removed, and the new FRET assay with positive and negative control were performed (Extended Data Fig. 9).

Extended Data Fig. 9

Minor comments

1) One Page 11, line 256 “homotypic” should be “homotypic”, and line 261 “pla” is a typo.

Response: We are very grateful for the careful review of the Referee. We have corrected the mistakes in the revised manuscript (line 276).

2) One Page 10, line 256 of Extended data. “efficiency” should be “efficiency”.

Response: We are very sorry for this obvious mistake, and we have corrected the spelling error in the revised manuscript (Extended Data Set, line 252).

Reviewers' Comments:

Reviewer #1:

Remarks to the Author:

The authors have addressed my concerns.

Reviewer #2:

Remarks to the Author:

In this revised manuscript, the authors have addressed my concerns. I have no additional issues to raise.

Reviewer #4:

Remarks to the Author:

The authors have satisfactorily addressed my questions.

REVIEWERS' COMMENTS

Reviewer #1 (Remarks to the Author):

The authors have addressed my concerns.

Reviewer #2 (Remarks to the Author):

In this revised manuscript, the authors have addressed my concerns. I have no additional issues to raise.

Reviewer #4 (Remarks to the Author):

The authors have satisfactorily addressed my questions.

Response: We are very grateful for the careful and helpful review of the Referees and are pleased that we could address all concerns satisfactorily.